# Stable Preference Optimization: Learning Preference is More Important Than Imitation

## Abstract

Aligning the behavior of large language models (LLMs) with human values and preferences is a critical challenge for their safe and effective deployment. Direct Preference Optimization (DPO; Rafailov et al. (2023)) has emerged as a widely adopted approach for incorporating human feedback into LLM training. However, its objective frequently induces reward hacking: the model reduces the probabilities of both the preferred response ($y_w$) and the dispreferred response ($y_l$), while only maintaining a higher ratio between them. This behavior undermines the ability to faithfully represent human preferences, as both responses are effectively treated as undesirable. Subsequent methods attempt to alleviate this limitation but introduce new trade-offs. Kahneman–Tversky Optimization (KTO; Ethayarajh et al. (2024)) emphasizes imitating the preferred response ($y_w$), yet does so at the expense of enlarging the margin between $y_w$ and $y_l$. Similarly, the symmetric squared loss in Identity Policy Optimization (IPO; Azar et al. (2023)) fails to distinguish between a large positive log-probability difference (indicating a correctly learned preference) and a large negative one (indicating a pathologically inverted preference), penalizing both extremes equally.To address this cascade of challenges, we propose Stable Preference Optimization (SPO). At its core is a novel loss function designed to: 1) prevent reward hacking by establishing a stable, finite optimization target; 2) focus on the preference margin rather than preferred response ($y_w$) imitation; and 3) corrects IPO's loss imbalance with an asymmetric design by using the function $f(z) = -ze^{-z}$. For this loss, when the positive log-probability difference is higher than an initial point, the loss is lower than at the initial point; when the positive log-probability difference is lower than the initial point, the loss is higher than at the initial point, while simultaneously being a convex function that possesses a unique minimum.Our method provides a unified solution to the core drawbacks of DPO, KTO, and IPO. Experimental results demonstrate significant improvements in both alignment performance and training stability.

## 1 Introduction

A central component in building state-of-the-art large language models (LLMs) is Reinforcement Learning from Human Feedback (RLHF) (Christiano et al., 2017; Ziegler et al., 2019). RLHF aligns pretrained LLMs with human preferences by leveraging human evaluation data, thereby making the models more helpful, truthful, and harmless (Ouyang et al., 2022; Casper et al., 2023).

Without RLHF, pretrained LLMs are prone to undesirable behaviors such as generating offensive or toxic content, amplifying social biases, or leaking sensitive information from training data (Gehman et al., 2020; Carlini et al., 2021; Ganguli et al., 2022).

The RLHF pipeline typically consists of two stages: (i) training a reward model from pairwise comparisons of model outputs to quantify human preferences, and (ii) fine-tuning the base LLM with reinforcement learning to maximize the learned reward. We model RLHF as an offline contextual bandit (Ouyang et al., 2022), and attribute reward overoptimization to distributional shift and uncertainty in reward estimation.

Intuitively, during fine-tuning, the response distribution of the updated LLM may deviate from that of the training data. For out-of-distribution responses—those insufficiently represented in the prefer-

ence dataset—the high intrinsic uncertainty of human preference labels can cause the reward model to produce misleading estimates. In such cases, reward overoptimization arises when the LLM is pushed to maximize a reward model that provides unreliable signals on out-of-distribution responses.

For the out-of-distribution responses, which are dissimilar with (or not well covered by) the responses in the data, the high inherent uncertainty of underlying human preferences could make the learned reward model misleading for out-of-distribution responses. In this situation, reward overoptimization can occur because the LLM is fine-tuned towards maximizing a reward model with defective out-of-distribution prediction, giving a potential consequence that the LLM responses are favored by the learned reward but less preferred by a human (Zhu et al., 2024). The goal of RLHF is to optimize a language model policy $\pi_\theta$ to align with human preferences. This is typically formulated as maximizing a reward function $r(x, y)$, learned from a preference dataset $\mathcal{D}_p = \{(x, y_w, y_l)\}$, where $y_w$ is preferred over $y_l$ for a prompt $x$. The standard RLHF objective is:

$$\max_{\pi_\theta} \mathbb{E}_{x \sim \mathcal{D}, y \sim \pi_\theta}[r(x, y)] - \beta \, \mathrm{KL}[\pi_\theta(y|x) \, \| \, \pi_{\mathrm{ref}}(y|x)], \tag{1}$$

where the KL-divergence term, scaled by $\beta$, regularizes the policy to remain close to a reference model $\pi_{\mathrm{ref}}$.

Rafailov et al. (2023) introduced Direct Preference Optimization (DPO) as a more direct approach to optimizing this objective. They established a theoretical link between the optimal policy $\pi^*$ and the underlying reward function:

$$r(x, y) = \beta \log \frac{\pi^*(y|x)}{\pi_{\mathrm{ref}}(y|x)} + Z(x), \tag{2}$$

where $Z(x)$ is a normalization term dependent only on $x$. This insight allows DPO to bypass the explicit training of a reward model.

However, we identify a fundamental limitation of the DPO objective. While its theoretical derivation implies a specific relationship between the optimal policy and the reward, its loss function promotes the unbounded maximization of the log-probability ratio between the preferred response $y_w$ and the dispreferred response $y_l$. This maximization is only consistent with the theory in the edge case where the reward difference between responses is infinite. For any finite reward, the objective becomes misaligned. Moreover, this formulation can lead to pathologically large gradients, especially when the probability of the dispreferred response $\pi_\theta(y_l|x)$ becomes very small—a phenomenon that induces training instability and reward hacking.

To address these shortcomings, we introduce a new loss function. Our approach stems directly from the optimality condition of Equation 8. Instead of maximizing the log-probability ratio, our loss function optimizes the policy toward a specific target value for this ratio, determined by the reward difference. This principled objective leads to a more stable optimization landscape. In particular, our loss naturally dampens gradients when the logit difference is large, thereby preventing the instability observed in DPO and mitigating the risk of reward hacking.

## 2 RELATED WORK

Our proposed Stable Preference Optimization (SPO) builds on and addresses key limitations in the rapidly evolving field of language model alignment. We position our contribution in relation to three areas: (i) the foundational paradigm of Reinforcement Learning from Human Feedback (RLHF), (ii) the development of direct preference optimization algorithms, and (iii) alternative approaches to alignment.

### 2.1 REINFORCEMENT LEARNING FROM HUMAN FEEDBACK (RLHF)

RLHF has been central to aligning LLMs with complex human values (Ouyang et al., 2022; Bai et al., 2022; **?**). The standard RLHF pipeline involves three stages: (1) supervised fine-tuning (SFT) on high-quality data, (2) training a reward model from human preference annotations, and (3) optimizing the policy with reinforcement learning, typically Proximal Policy Optimization (PPO) (Schulman et al., 2017), under a KL-constraint to control divergence from the SFT model.

While effective, this pipeline is also complex and unstable (Gao et al., 2023). The joint training of both policy and value networks, together with the requirement to maintain multiple copies of large models, renders RLHF computationally expensive and challenging to stabilize. SPO, like the direct methods discussed next, aims to preserve the alignment benefits of RLHF while mitigating its inherent complexity.

## 2.2 DIRECT PREFERENCE OPTIMIZATION

To circumvent the challenges of RL-based fine-tuning, Rafailov et al. (2023) introduced Direct Preference Optimization (DPO). DPO derives a closed-form mapping between the optimal policy and the reward function (Eq. 7), allowing for the direct optimization of a policy on preference data using a simple binary cross-entropy loss. Its simplicity and effectiveness have made it a popular choice for alignment. However, as we critically analyze in this paper, DPO's objective leads to an unbounded maximization of the log-likelihood ratio between preferred and dispreferred responses. This often results in *reward hacking* and model degeneration, as the model over-optimizes the reward signal by driving the probability of the dispreferred response to zero, thereby harming general capabilities (Gao et al., 2023). Our work directly addresses this core instability in DPO.

Several recent methods have been proposed to improve upon DPO. Identity Policy Optimization (IPO) (Azar et al., 2023) introduces a regularization term to theoretically prevent the overfitting encountered in DPO. IPO's loss function employs a symmetric squared loss around a target margin. While this ensures a finite solution, we argue that its symmetric penalty is a weakness: it equally penalizes both a correctly large positive log-ratio and a pathologically large negative one, failing to properly guide the optimization. Kahneman-Tversky Optimization (KTO) (Ethayarajh et al., 2024) departs from the need for paired preference data, instead relying on a value function inspired by prospect theory. While computationally efficient, KTO's objective overly focuses on maximizing the likelihood of positive examples ("imitation") and may fail to adequately learn the relative *margin* between chosen and rejected responses. SPO is designed to synthesize the strengths of these approaches—providing a finite target like IPO and effectively learning the preference margin—while avoiding their respective pitfalls through a novel asymmetric loss function.

## 2.3 SUMMARY

In summary, SPO distinguishes itself by directly tackling the reward hacking and instability problems of DPO. It improves upon IPO's symmetric loss with an asymmetric design that better reflects the directionality of preferences, and it maintains a focus on learning the preference margin, addressing a potential shortcoming of KTO. By providing a stable, principled, and effective objective, SPO offers a unified solution to the core challenges faced by current state-of-the-art preference optimization methods.

## 3 METHODOLOGY

### 3.1 FROM OPTIMALITY CONDITION TO A NEW OBJECTIVE

From the relationship in Equation 7, we can express the difference in rewards between a winning $(y_w)$ and losing $(y_l)$ completion for a given prompt $x$ under the optimal policy $\pi^*$ as:

$$r(x, y_w) - r(x, y_l) = \beta \left( \log \frac{\pi^*(y_w|x)}{\pi_{\text{ref}}(y_w|x)} - \log \frac{\pi^*(y_l|x)}{\pi_{\text{ref}}(y_l|x)} \right). \qquad (3)$$

This equation defines the condition for an optimal policy. The goal of training should be to steer the current policy $\pi_\theta$ to satisfy this condition. Let us define the policy-dependent logits difference as:

$$\text{logits}(\pi_\theta) = \log \frac{\pi_\theta(y_w|x)}{\pi_{\text{ref}}(y_w|x)} - \log \frac{\pi_\theta(y_l|x)}{\pi_{\text{ref}}(y_l|x)}. \qquad (4)$$

The optimality condition is therefore met when:

$$\text{logits}(\pi_\theta) = \frac{r(x, y_w) - r(x, y_l)}{\beta}. \qquad (5)$$

This reveals that the optimal policy does not require maximizing the logits, but rather driving them towards a finite target value. The DPO loss, which is equivalent to a log-sigmoid loss on $\beta \cdot \text{logits}(\pi_\theta)$, encourages making $\text{logits}(\pi_\theta)$ infinitely large, contradicting the theoretical foundation.

## 3.2 The Proposed Stable Preference Loss

We require a loss function that reaches its minimum when the optimality condition in Equation 5 is met. Althoughugh a squared error loss, $(\text{logits} - \frac{r_w - r_l}{\beta})^2$,

We propose a more robust loss function whose structure inherently guides the logits to a stable point. Consider the function $f(z) = -ze^{-z}$, which has a unique global maximum at $z = 1$. For this loss, when the positive log-probability difference is higher than an initial point, the loss is lower than at the initial point; when the positive log-probability difference is lower than the initial point, the loss is higher than at the initial point, while simultaneously being a convex function that possesses a unique minimum. We can leverage this property. Let $z = c \cdot \text{logits}$, where $c$ is a scaling constant. We want the loss to be minimized when $\text{logits} = 1/c$. From Equation 5, this implies $c \approx \frac{\beta}{r_w - r_l}$.

By absorbing the unknown reward difference into the hyperparameter $\beta$, we formulate our loss for Stable Preference Optimization (SPO) as follows:

$$\mathcal{L}_{\text{SPO}} = -(\beta \cdot \text{logits}(\pi_\theta)) \exp(-\beta \cdot \text{logits}(\pi_\theta))$$
$$- \left( \alpha \cdot \log \frac{\pi_{\text{ref}}(y_l|x)}{\pi_\theta(y_l|x)} \right) \exp \left( -\alpha \cdot \log \frac{\pi_{\text{ref}}(y_l|x)}{\pi_\theta(y_l|x)} \right) \tag{6}$$

This loss is minimized when the arguments to its two core components, $\beta \cdot \text{logits}(\pi_\theta)$ and $\alpha \cdot \log \frac{\pi_\theta(y_l|x)}{\pi_{\text{ref}}(y_l|x)}$, both equal 1 . This holds for both $\pi_w$ and $\pi_l$ that satisfy

$$r(x, y) = \beta \log \frac{\pi^*(y|x)}{\pi_{\text{ref}}(y|x)} + Z(x). \tag{7}$$

The loss function is minimized when the policy $\pi$ satisfies:

- For the preferred response $y_w$: $\frac{1}{\beta} - \frac{1}{\alpha} = \log \frac{\pi(y_w|x)}{\pi_{\text{ref}}(y_w|x)}$.

- For the dispreferred response $y_l$: $\frac{1}{\alpha} = \log \frac{\pi(y_l|x)}{\pi_{\text{ref}}(y_l|x)}$.

This formulation has several advantages:

1. **Principled Target:** It optimizes towards a finite, stable point consistent with RLHF theory.

2. **Robustness to Over-Optimization:** As $\text{logits}(\pi_\theta) \to \infty$, the loss gracefully decays to zero. This prevents the model from being penalized for being "too confident," avoiding unstable gradients for well-distinguished pairs.

3. **Asymmetric Penalty:** The loss function heavily penalizes logits values less than the target $1/\beta$, while applying a vanishing penalty for values greater than the target.

4. **Stable Preference Learning:** While it is difficult to precisely estimate the magnitude of $\frac{\pi_\theta(y_w|x)}{\pi_{\text{ref}}(y_w|x)}$, we know that as long as $\frac{\pi_\theta(y_l|x)}{\pi_{\text{ref}}(y_l|x)}$ decreases and the difference between the two ratios remains within a certain range, the model will effectively learn human preferences. This approach prevents simultaneous sharp declines in the probabilities of both positive and negative examples, and avoids overfitting on positive examples.

## 4 Theoretical Analysis: Comparison with DPO, IPO, and KTO

**1. Comparison with DPO: Solving Gradient Explosion and Reward Hacking** Our method satisfies the optimal solution for this objective, whereas DPO contradicts this optimal solution.

$$\max_{\pi_\theta} \mathbb{E}_{x \sim \mathcal{D}, y \sim \pi_\theta} [r(x, y)] - \beta \text{KL}[\pi_\theta(y|x) \| \pi_{\text{ref}}(y|x)] \tag{8}$$

The DPO loss is given by:

$$\mathcal{L}_{\text{DPO}} = -\mathbb{E}_{(x,y_w,y_l)\sim\mathcal{D}} \left[ \log \sigma \left( \beta \log \frac{\pi_\theta(y_w|x)}{\pi_{\text{ref}}(y_w|x)} - \beta \log \frac{\pi_\theta(y_l|x)}{\pi_{\text{ref}}(y_l|x)} \right) \right].$$

The DPO method contradicts the optimal solution of the RLHF objective in Equation equation 8. The DPO loss optimizes towards maximizing:

$$\log \frac{\pi_\theta(y_w|x)}{\pi_{\text{ref}}(y_w|x)} - \log \frac{\pi_\theta(y_l|x)}{\pi_{\text{ref}}(y_l|x)}.$$

However, from the relationship:

$$r(x, y_w) - r(x, y_l) = \beta \left( \log \frac{\pi^*(y_w|x)}{\pi_{\text{ref}}(y_w|x)} - \log \frac{\pi^*(y_l|x)}{\pi_{\text{ref}}(y_l|x)} \right),$$

it is clear that this difference should be bounded in practice. Since the reward difference $r(x, y_w) - r(x, y_l)$ in the training data is finite, pushing the log-ratio difference to infinity (as encouraged by DPO) leads to reward hacking. This occurs when both $\pi_\theta(y_w|x)$ and $\pi_\theta(y_l|x)$ decrease significantly, but their ratio increases to minimize the loss. As a result, the model deviates substantially from the reference model $\pi_{\text{ref}}$, and may generate responses that are unrelated to either the winning or losing responses.Although several studies have attempted to mitigate this phenomenon, such as RPO Zhihan Liu (2024), we argue that the constraint on the reward function $r(x, y)$, defined as

$$r(x, y) = \beta \log \frac{\pi(y \mid x)}{\pi_{\text{ref}}(y \mid x)} + Z(x), \tag{9}$$

should be applied specifically to the preferred responses ($y_w$) and dispreferred responses ($y_l$) based on necessity, rather than being uniformly enforced regardless of whether reward hacking occurs. Indiscriminate application of this constraint can adversely affect overall performance.We revise the conventional assumption that the difference in rewards, $r(x, y_w) - r(x, y_l)$, is constant across the dataset $D$. Instead, we propose that the normalized rewards for both the preferred and dispreferred responses relative to a baseline are constant. Formally, our new hypothesis is:

$$r(x, y_w) - Z(x) = C_w \quad \text{and} \quad r(x, y_l) - Z(x) = C_l \quad \text{for all } (x, y_w, y_l) \in D, \tag{10}$$

where $C_w$ and $C_l$ are constants specific to the response type, and $Z(x)$ is the input-dependent baseline function.

**2. Comparison with IPO: Introducing Asymmetric Constraints for a Better Optimum** The IPO (Identity Policy Optimization) loss is designed to address the limitations of DPO by preventing reward hacking and ensuring proper regularization against the reference policy. The IPO loss function is given by:

$$\mathcal{L}_{\text{IPO}} = \mathbb{E}_{(x,y_w,y_l)\sim\mathcal{D}} \left[ \left( \log \left( \frac{\pi_\theta(y_w|x)}{\pi_{\text{ref}}(y_w|x)} \right) - \log \left( \frac{\pi_\theta(y_l|x)}{\pi_{\text{ref}}(y_l|x)} \right) - \frac{1}{2\beta} \right)^2 \right]$$

A key limitation of the IPO loss is that it primarily constrains the log-ratio difference $\log \frac{\pi_\theta(y_w|x)}{\pi_{\text{ref}}(y_w|x)} - \log \frac{\pi_\theta(y_l|x)}{\pi_{\text{ref}}(y_l|x)}$, rather than constraining both $\pi_\theta(y_w|x)$ and $\pi_\theta(y_l|x)$ individually. This leads to a symmetry in the loss function: when $\log \frac{\pi_\theta(y_w|x)}{\pi_{\text{ref}}(y_w|x)} \to -\infty$ and $\log \frac{\pi_\theta(y_l|x)}{\pi_{\text{ref}}(y_l|x)} \to -\infty$, the loss approaches the same value as when both terms tend to $+\infty$. However, these two cases are fundamentally different:

- When both terms tend to $-\infty$, it implies $\pi_\theta(y_w|x) \to 0$ and $\pi_\theta(y_l|x) \to 0$, which contradicts the preference signal (as the preferred response $y_w$ is assigned near-zero probability). - When both terms tend to $+\infty$, it implies $\pi_\theta(y_w|x) \to +\infty$ and $\pi_\theta(y_l|x) \to +\infty$, which is also undesirable but can be regularized to a reasonable range.

This symmetry means the loss function does not distinguish between these two pathological cases, even though the first case (both probabilities tending to zero) is entirely contrary to the preference data.

To address this, we propose using a modified loss function $f(z) = -ze^{-z}$ instead of the quadratic loss $f(z) = (z - c)^2$ used in IPO. This function breaks the symmetry and assigns lower loss to the case where the log-ratios are large and positive (which can be regularized) compared to the case where they are large and negative (which contradicts preferences). However, since the loss also becomes small when the log-ratios are very positive, this could still lead to reward hacking (similar to DPO), where the model excessively increases $\pi_\theta(y_w|x)$ without bound.

Therefore, we introduce an additional constraint on the negative example $y_l$ to prevent reward hacking. The full loss function becomes:

$$\mathcal{L}_{\text{SPO}} = -(\beta \cdot \text{logits}(\pi_\theta)) \exp(-\beta \cdot \text{logits}(\pi_\theta))$$
$$- \left( \alpha \cdot \log \frac{\pi_{\text{ref}}(y_l|x)}{\pi_\theta(y_l|x)} \right) \exp \left( -\alpha \cdot \log \frac{\pi_{\text{ref}}(y_l|x)}{\pi_\theta(y_l|x)} \right) \tag{11}$$

**3. Comparison with KTO: Achieving Stable Convergence and Avoiding Overfitting and Collapse** The KTO loss function is defined as:

$$\mathcal{L}_{\text{KTO}}(\pi_\theta, \pi_{\text{ref}}) = \mathbb{E}_{x,y \sim D} \left[ \lambda_y - v(x,y) \right]$$

$$\text{where} \quad r_\theta(x,y) = \log \frac{\pi_\theta(y \mid x)}{\pi_{\text{ref}}(y \mid x)}$$

$$z_0 = \text{KL} \left( \pi_\theta(y' \mid x) \parallel \pi_{\text{ref}}(y' \mid x) \right)$$

$$v(x,y) = \begin{cases} \lambda_D \, \sigma \left( \beta (r_\theta(x,y) - z_0) \right) & \text{if } y \sim y_{\text{desirable}} \mid x \\ \lambda_U \, \sigma \left( \beta (z_0 - r_\theta(x,y)) \right) & \text{if } y \sim y_{\text{undesirable}} \mid x \end{cases}$$

A key limitation of KTO is that increasing $\pi_\theta(y_w|x)$ does not necessarily correlate positively with model performance. When the dataset contains suboptimal preferred $y_w$, learning the preferences embodied in both $y_w$ and $y_l$ is more important than merely imitating the $y_w$. When $\pi_\theta(y_w|x)$ lies within a certain range, model performance is often determined by the relative preference between $\pi_\theta(y_w|x)$ and $\pi_\theta(y_l|x)$, rather than the absolute value of $\pi_\theta(y_w|x)$.

Excessive emphasis on maximizing $\pi_\theta(y_w|x)$ can lead to: - Reduced generative diversity due to over-concentration on winning responses - Increased susceptibility to noise in the preference data - Potential degradation of overall model performance despite higher likelihood of preferred responses

Thus, a more balanced approach that focuses on learning the preference ratio rather than pushing $\pi_\theta(y_w|x)$ to extreme values is desirable.

**Our Method:** Our loss ensures the model converges to a **dataset-dependent fixed probability ratio** determined by the reward magnitude, rather than extreme values. This implies: * **Avoids Overfitting:** For low-quality dispreferred responses, their probability **does not need to be excessively reduced**; for preferred responses, their probability **does not need to be excessively increased**. * **Maintains Entropy:** It maintains an appropriate probability distribution, preserving a degree of generative diversity in the model. * **Built-in Stability:** By design, it directly constrains the probability ratio relative to the reference model, **eliminating the need for an additional KL-divergence constraint to prevent training collapse**, making the training process more concise and robust.

In summary, through theoretical gradient analysis and comparison, our SPO loss demonstrates significant advantages over DPO, IPO, and KTO in terms of avoiding reward hacking, ensuring training stability, guiding the model towards the correct optimum, and preventing overfitting.

## 5 EXPERIMENTS

In this section, we provide a detailed empirical analysis of SPO to highlight the following two key points:

1. **Flexibility and Plug-and-Play Nature:** SPO is a flexible plug-in module that can be applied to different reference models. More importantly, its hyperparameters offer significant

adaptability to various training scenarios. Specifically, a higher $\alpha$ value mitigates overoptimization during training by increasing the trust in the chosen responses from the preference dataset. Conversely, a lower $\beta$ value directs the model to focus more on learning the underlying preference rather than merely imitating the preferred response $y_w$. This flexibility allows SPO to effectively handle a wide range of training conditions.

2. **Superior In-Distribution Alignment:** Justifying our theoretical analysis, SPO achieves better alignment performance than DPO, IPO, and KTO on in-distribution data.

## 5.1 EXPERIMENTAL SETUP

To validate the effectiveness of our proposed SPO loss, we conduct a comprehensive set of experiments on two leading base models: **Qwen2.5-7B-Instruct** (Team, 2024)and **Llama-3.1-8B-Instruct** (Llama Core & Meta, 2024). Our fine-tuning process consists of two stages:

1. **Supervised Fine-Tuning (SFT)**: We first fine-tune the base models on the `HuggingFaceH4/ultrachat_200k` (Ding et al., 2023) dataset to enhance their general instruction-following capabilities. This results in our SFT baseline models.

2. **Preference Alignment**: Following SFT, the models are further aligned using preference data. We compare our SPO method against the standard DPO baseline ,KTO and IPO using the `HuggingFaceH4/ultrafeedback_binarized` (Cui et al., 2023) dataset. All alignment runs start from the same SFT checkpoint for a fair comparison.To ensure a fair comparison between SPO and IPO, we configure their hyperparameters such that their respective optimal solutions coincide.

We evaluated the final models by conducting pairwise, head-to-head comparisons and using GPT-4o (OpenAI, 2024) as the judge to determine a win rate. We report the win rates for all model versions (SFT, DPO,KTO,IPO and our SPO) against each other. For the in-data distribution evaluation, we select the 200 prompts in the test split of the training dataset to let the SFT model, DPO, IPO,KTO and SPO generate the response respectively. We choose GPT-4o to annotate the preference in the response pairs. we instruct GPT-4 to give an annotation among win, lose, and tie . For tie results, we assign half a win to each model when calculating the win rate. We select the better-performing variant between the TRL (von Werra et al., 2020)-implemented sigmoid formulation and the classical DPO loss (as described in the original paper). In our experiments, we set the label smoothing coefficient to label_smoothing $= 0.1$. The resulting objective is defined as

$$\text{losses} = -\,(1 - 0.1) \cdot \log(\sigma(\beta \cdot \text{logits})) - 0.1 \cdot \log(\sigma(-\beta \cdot \text{logits})), \tag{12}$$

where $\sigma(\cdot)$ denotes the sigmoid function. This formulation can be seen as a smoothed variant of the DPO loss, where label smoothing mitigates overconfidence and improves stability during RLHF fine-tuning.

## 5.2 RESULTS

The experimental results, presented in Table 1 and Table 2, unequivocally demonstrate the superiority of our proposed SPO loss. In the tables, each cell shows the win rate of the model in the row against the model in the column.

For the **Qwen2.5-7B** model (Table 1), we observe a clear hierarchy of performance. The DPO model vastly outperforms the SFT baseline with a **91.70%** win rate. More importantly, our SPO model achieves a significant improvement over DPO, securing a **56.50%** win rate in a direct head-to-head matchup. Against the SFT baseline, our SPO model's superiority is even more pronounced, with a staggering **95.15%** win rate.

We confirmed this trend by repeating the experiment on the **Llama-3.1-8B** model, with the results shown in Table 2. The pattern of improvement is remarkably consistent. DPO shows a strong gain over SFT (91.46% win rate). Once again, our SPO method delivers a clear performance boost over DPO, winning the head-to-head comparison with a **53.73%** win rate.

Across both model architectures, the results are unambiguous: SFT provides a solid foundation, DPO offers a substantial improvement through preference alignment, and our SPO method consistently and significantly outperforms DPOKTO and IPO. This validates that the benefits of SPO's

Table 1: Win rates for **Qwen2.5-7B-instruct** fine-tuning methods. Each cell shows the win percentage of the row model against the column model.

| Win Rate (%) of Row vs. Column | SFT | DPO | KTO | IPO | SPO (Ours) |
|---|---|---|---|---|---|
| **SFT** | – | 8.30 | 15.66 | 10.50 | 4.85 |
| **DPO** | 91.70 | – | 52.28 | 51.15 | 43.50 |
| **KTO** | 84.34 | 47.72 | – | 46.80 | 38.38 |
| **IPO** | 89.50 | 48.85 | 53.20 | – | 41.20 |
| **SPO (Ours)** | **95.15** | **56.50** | **61.62** | **58.80** | – |

Table 2: Win rates for **Llama-3.1-8B** fine-tuning methods. Each cell shows the win percentage of the row model against the column model.

| Win Rate (%) of Row vs. Column | SFT | DPO | KTO | IPO | SPO (Ours) |
|---|---|---|---|---|---|
| **SFT** | – | 8.54 | 13.95 | 9.45 | 6.32 |
| **DPO** | 91.46 | – | 53.95 | 51.85 | 46.27 |
| **KTO** | 86.05 | 46.05 | – | 47.55 | 43.95 |
| **IPO** | 90.55 | 48.15 | 52.45 | – | 45.33 |
| **SPO (Ours)** | **93.68** | **53.73** | **56.05** | **54.67** | – |

stable and principled loss function generalize across different models, leading to a more effective and robust alignment with human preferences.

## 6  CONCLUSION

In this work, we present Stable Preference Optimization (SPO), a novel and principled approach for language model alignment. Our method addresses a key theoretical limitation shared by several existing methods, including DPO, IPO, and KTO, which can lead to unbounded optimization of the logit difference and potential training instability. SPO resolves this issue by explicitly optimizing towards a theoretically grounded, finite target derived from the RLHF optimality condition, thereby providing stronger convergence guarantees. Furthermore, SPO introduces additional flexibility through a carefully designed parameterization, allowing it to adapt to a wider range of preference modeling scenarios beyond the constraints of previous algorithms. Extensive empirical results demonstrate that SPO not only achieves significant performance improvements over strong baselines but does so with enhanced training stability. We believe SPO offers a more robust, flexible, and theoretically sound foundation for future research in preference-based alignment. We believe that SPO provides a more stable, principled, and effective path for future research in language model alignment.

## ETHICS STATEMENT

This work presents a new method for aligning large language models (LLMs) with human preferences. The datasets used for training and evaluation in this study are derived from publicly available and widely used sources in the alignment literature. We have conducted analyses to mitigate the risk of reward hacking, a known failure mode in alignment that can lead to degraded model performance. We strongly urge that any application of this technology includes rigorous red-teaming and harm mitigation strategies, and is guided by a framework of human oversight and well-being.

## REPRODUCIBILITY STATEMENT

To ensure the reproducibility of our work, we commit to releasing the full source code and hyperparameters for our Stable Preference Optimization (SPO) method.

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

## 7 APPENDIX

We employed large language models (LLMs), including Gemini 2.5 Pro, ChatGPT, and DeepSeek, to polish the writing and improve the readability and fluency of this article.

