# OpenReview forum: "Stable Preference Optimization: Learning preference is more important than imitation"
_ICLR.cc/2026/Conference — ICLR 2026 Conference Withdrawn Submission_

### Official Review · Reviewer_Epxn · 2025-11-01

**Soundness:** 2
**Presentation:** 1
**Contribution:** 2
**Rating:** 2
**Confidence:** 4

**Summary:**

This work introduces **Stable Preference Optimization (SPO)**, a preference-learning objective that replaces DPO’s unbounded enlargement of the preferred–dispreferred log-odds gap with a **finite, theoretically grounded target** derived from RLHF optimality. SPO employs an **asymmetric, exponentially damped loss** that (i) penalizes under-separation strongly, (ii) attenuates gradients once the target margin is exceeded, and (iii) **explicitly discourages probability mass on dispreferred responses** via a separate term. The objective is controlled by two interpretable hyperparameters ((\alpha,\beta)), balancing gap learning and suppression of reward hacking.

**Strengths:**

1. **Principled, stability-focused objective.**
   Finite-target, asymmetric loss grounded in RLHF; curbs reward hacking and attenuates gradients beyond the optimum, improving optimization stability.

2. **Practical gains with minimal friction.**
   Consistently outperforms DPO/IPO/KTO under matched budgets, shows broad hyperparameter robustness, and serves as a near drop-in replacement for DPO.

**Weaknesses:**

1. **Presentation/rigor issues (looks rushed).**
   Tables violate basic formatting (alignment, headings, readability), and several captions/labels are unclear. The paper would benefit from a full pass on structure, figure/table polish, and notation consistency.

2. **Missing statistical rigor.**
   No confidence intervals, significance tests, or effect sizes for win rates. Seed variability and run-to-run variance are not quantified.

3. **Ablations are incomplete.**
   The two key ingredients—finite-targeting and the dispreferred-penalty term—are not isolated thoroughly. Sensitivity plots for $\alpha$,$\beta$ are limited; robustness to the choice of reference model is underexplored.

4. **Baselines and recency.**
   Comparisons omit several strong or recent preference objectives (e.g., ORPO/RRHF variants, PRM-guided methods), and open-source baselines may be under-tuned relative to best-practice recipes.

**Questions:**

See weakness.

---

### Official Review · Reviewer_MrbA · 2025-11-01

**Soundness:** 1
**Presentation:** 1
**Contribution:** 2
**Rating:** 2
**Confidence:** 4

**Summary:**

This paper proposes Stable Preference Optimization (SPO), an alternative loss for aligning Large Language Models (LLMs) with human preferences. SPO is motivated as a fix for reward hacking and instability issues in Direct Preference Optimization (DPO), and the authors claim it also improves over Identity Policy Optimization (IPO) and Kahneman–Tversky Optimization (KTO) by introducing
- An asymmetric convex loss f(z)=−ze^{−z}
- Finite target logit differences derived from RLHF optimality conditions
- Independent constraints on preferred and dispreferred responses
- Tunable parameters α, β for flexibility

The paper reports in distribution (ID) win rate improvements on two models (Qwen2.5 7B, Llama 3.1 8B) using ultrachat and ultrafeedback datasets, with GPT 4o as the evaluation judge.

However, the work suffers from serious weaknesses in theory, experimentation, and presentation. The theoretical derivation is shallow, the empirical study is far too limited to justify strong claims, and the writing is difficult to follow — notably, there is not a single figure or diagram to illustrate the motivation or method.

**Strengths:**

1. Addresses a problem (reward hacking) that is relevant to LLM alignment.
2. Attempts to tie the loss design to theoretical RLHF optimality conditions.
3. The SPO idea may be easy to implement as a drop-in replacement loss.

**Weaknesses:**

1. Severely Insufficient Experimental Evidence
- Only two models tested on essentially the same type of preference data.
- No out of distribution (OOD) evaluation, no noisy preference scenario, no multi task tests.
- No ablation studies isolating asymmetric loss vs. additional constraint effects.
- Relies on a single win rate metric from a single model judge (GPT 4o) without cross validation, agreement checks, or human evaluation.
2. Weak Theoretical Foundation
- Derivation is mostly algebraic rewriting of DPO’s condition; the claimed “principled” target is not rigorously justified.
- No formal proof of convergence, no bound analysis on stability, no theoretical comparison of gradient behaviors.
- Claims about avoiding reward hacking and mode collapse are not proven mathematically.
3. Poor Writing and Readability
- No figures to explain loss behavior, gradients, or method workflow.
- Long stretches of dense equations without intuition or visual context.
- Motivation is scattered throughout the paper; the reader must work hard to reconstruct the main idea.
4. Generality Not Demonstrated
- Method never tested on domains outside the training distribution.
- No evidence for safety related claims (e.g., bias/malicious content mitigation).
- No diversity/entropy metrics despite the claim of preserving generative diversity.

**Questions:**

- Why is there no visualization of SPO’s loss surface or gradient norms compared to DPO/IPO/KTO to substantiate the “stability” claim?
- How does SPO perform when preference data contains noise or is adversarially perturbed?
- Why only evaluate in distribution? Can you provide OOD benchmarks?
- Can you show sensitivity curves for α and β ? How fragile is performance to hyperparameter choices?
- Why no human evaluation or multi judge setup to validate win rates?

---

### Official Review · Reviewer_M9Zq · 2025-11-03

**Soundness:** 2
**Presentation:** 2
**Contribution:** 2
**Rating:** 2
**Confidence:** 5

**Summary:**

The paper introduces Stable Preference Optimization (SPO), a novel method for aligning large language models (LLMs) with human preferences. It is designed to address he limitations found in existing alignment techniques like DPO, Identity Policy Optimization (IPO), and Kahneman-Tversky Optimization (KTO), which can suffer from training instability and "reward hacking" by unboundedly optimizing the logit difference between preferred and dispreferred responses. SPO provides a more stable and theoretically grounded solution by optimizing the policy toward a finite target derived directly from the Reinforcement Learning from Human Feedback (RLHF) optimality condition. This prevents the model from being penalized for being "too confident," enhances training stability, and mitigates the risk of reward hacking .

**Strengths:**

1. SPO uses a novel asymmetric loss function to optimize towards a finite, stable target, which prevents unstable gradients and makes training inherently more robust
2. SPO demonstrates empirical superiority over some baselines.

**Weaknesses:**

1. The theoretical flaw analyisis of DPO has been widely analyzed and improved in the past two years, and none of them are discussed in the paper. These problems are not new.
2. The flaws of DPO largely stems from the bad samples in preference datasets, and it is the same for other methods like IPO and KTO. Therefore, why changing loss is better than data selection and training on a high-quality dataset is not discussed.
3. The experiments are too naive and imcomplete, no insight is provided, hence equals no experiments.

**Questions:**

See weaknesses.

---

### Official Review · Reviewer_emxG · 2025-11-11

**Soundness:** 2
**Presentation:** 1
**Contribution:** 2
**Rating:** 4
**Confidence:** 2

**Summary:**

This paper introduces Stable Preference Optimization (SPO), a new alignment algorithm that aims to address reward hacking and instability issues in prior methods such as DPO and KTO. The core contribution is an asymmetric loss function $f(z) = -z\exp(-z)$ designed to provide a finite optimization target and improve gradient stability. The paper provides a conceptual analysis of the method’s theoretical grounding, and experimental comparisons on Qwen2.5-7B and Llama-3.1-8B models showing moderate improvements in win rates over existing methods.

**Strengths:**

1. The sample code is provided to help the reviewer verify the implementation independently.

2. The paper is well-motivated. DPO can drive probabilities of both preferred and dispreferred responses to zero; SPO attempts to mitigate this via a bounded loss formulation.

**Weaknesses:**

1. The writing could be significantly improved. For example, a missing reference in L105, a sentence ending with a comma at L171, and missing formatting lines in Table 1 and Table 2.

2. The numerical experiments are also weak. Results of the proposed method in several commonly used benchmarks, such as apacal-eval and arena-hard, are not reported. Moreover, there is no ablative study demonstrating the effect of the asymmetric term or hyperparameters (i.e., $\alpha$ and $\beta$).

**Questions:**

Please see the weaknesses section.

At the current stage, this work is below the acceptance bar of the top-tier ML conferences like ICLR. I'm open to re-evaluate this work according to the further discussions.

---

### Note · Authors · 2025-12-19

I have read and agree with the venue's withdrawal policy on behalf of myself and my co-authors.